# Study of the Correlation between the Amorphous Indium-Gallium-Zinc Oxide Film Quality and the Thin-Film Transistor Performance

**DOI:** 10.3390/nano11020522

**Published:** 2021-02-18

**Authors:** Shiben Hu, Kuankuan Lu, Honglong Ning, Rihui Yao, Yanfen Gong, Zhangxu Pan, Chan Guo, Jiantai Wang, Chao Pang, Zheng Gong, Junbiao Peng

**Affiliations:** 1Institute of Semiconductors, Guangdong Academy of Sciences, Guangzhou 510651, China; hushiben@foxmail.com (S.H.); gongyanfen@gdisit.com (Y.G.); panzhangxu@gdisit.com (Z.P.); guochan@gdisit.com (C.G.); wangjiantai@gdisit.com (J.W.); pangchao@gdisit.com (C.P.); 2Institute of Polymer Optoelectronic Materials and Devices, State Key Laboratory of Luminescent Materials and Devices, South China University of Technology, Guangzhou 510641, China; kk-lu@foxmail.com (K.L.); yaorihui@scut.edu.cn (R.Y.); psjbpeng@scut.edu.cn (J.P.)

**Keywords:** oxide semiconductor, a-IGZO, TFT, X-ray reflectivity, microwave photoconductivity decay

## Abstract

In this work, we performed a systematic study of the physical properties of amorphous Indium–Gallium–Zinc Oxide (a-IGZO) films prepared under various deposition pressures, O${}_{2}$/(Ar+O${}_{2}$) flow ratios, and annealing temperatures. X-ray reflectivity (XRR) and microwave photoconductivity decay (μ-PCD) measurements were conducted to evaluate the quality of a-IGZO films. The results showed that the process conditions have a substantial impact on the film densities and defect states, which in turn affect the performance of the final thin-film transistors (TFT) device. By optimizing the IGZO film deposition conditions, high-performance TFT was able to be demonstrated, with a saturation mobility of 8.4 cm${}^{2}$/Vs, a threshold voltage of 0.9 V, and a subthreshold swing of 0.16 V/dec.

## 1. Introduction

Amorphous Indium-Gallium-Zinc Oxide (a-IGZO) thin-film transistors (TFTs) have recently been considered as the most promising candidate for the new display backplane due to their high mobility, good uniformity, low off current, and good process compatibility with conventional a-Si TFTs [1,2,3]. It is well recognized that the performance of a-IGZO TFTs is governed by film qualities, which are sensitive to their process conditions [4,5,6,7]. In particular, the film density is a good indicator of the film quality [8,9] because a sparse film structure provides spaces where impurity species are incorporated, such as weakly bonded, interstitial, excess oxygen. To effectively evaluate the IGZO thin film quality, various methods have been established. Among them, the microwave photoconductivity decay (μ-PCD) method is a popular technique for the characterization of carrier trap defects in semiconductors, due to its advantage of the non-destructive and contactless measurement [10,11].

Hence, in the present study, we used X-ray reflectivity (XRR) and μ-PCD to evaluate the quality of the a-IGZO film prepared at various deposition O${}_{2}$ ratios, sputtering pressures, and annealing temperatures. We found that there is a strong correlation between the film quality and the device performance. Our systematic study indicates high-quality a-IGZO film with high densities and few traps can be formed under low O${}_{2}$/(Ar+O${}_{2}$) ratio, low sputtering pressure, and high-temperature annealing. Consequently, high-performance TFTs can be made with the high-quality a-IGZO film.

## 2. Materials and Methods

TFTs with an inverted staggered structure were fabricated in this study, as shown in Figure 1. A 300-nm-thick Al film was deposited on a glass substrate by DC magnetron sputtering and then patterned by wet etching to form the backside gate electrode. A 200-nm-thick Al${}_{2}$O${}_{3}$ gate insulator was then formed by an anodizing process [12]. After that, a 25-nm-thick a-IGZO (In:Ga:Zn = 1:1:1, at.%) layer was deposited on Al${}_{2}$O${}_{3}$ by radio frequency (RF) magnetron sputtering under various conditions. The deposition temperature and sputtering power were fixed at room temperature and 80 W. After the deposition, the a-IGZO films were annealed in ambient air for 1 h. In order to evaluate the influence of process conditions on the film quality, various parameters, including the O${}_{2}$ ratio, sputtering pressure, and post-deposition temperature, were evaluated according to the following reference conditions: 5% O${}_{2}$ ratio, 0.67 Pa sputtering pressure, 450 °C post-deposition temperature. Next, a 100-nm-thick Mo film was deposited on the a-IGZO film by DC magnetron sputtering. The DC sputtering power and working pressures were 100 W and 0.27 Pa, respectively. The IGZO and Mo were both patterned through shadow masks to form channel and source/drain (S/D) electrodes. The channel width and channel length were 625 μm and 500 μm, respectively.

The relevant film samples were measured by XRR (PANalytical Empyrean) and μ-PCD (KOBELCO, LTA-1620SP). The electrical characterization of the TFTs was performed using a semiconductor parameter analyzer (Agilent 4155C) under ambient conditions.

## 3. Results and Discussions

Figure 2 presents the XRR curves of a-IGZO films prepared under various process conditions. It is observed that O${}_{2}$/(Ar+O${}_{2}$) ratio and pressure significantly affect the film density, whereas the annealing temperature has a weak effect. As the O${}_{2}$/(Ar+O${}_{2}$) ratio and the sputtering pressure increased (0% to 70%, 0.13 Pa to 2.67 Pa), the critical angle of the XRR curves shifted towards a lower angle, indicating that a-IGZO film deposited at lower O${}_{2}$ ratio and pressure was denser than that at higher O${}_{2}$ ratio and pressure. The optimal density could be as high as 6.0 g/cm${}^{3}$ (0.67 Pa, 0% O${}_{2}$), which is very close to the crystalline IGZO film (6.4 g/cm${}^{3}$) [9]. And the lowest density of 5.3 g/cm${}^{3}$ (2.67 Pa, 5% O${}_{2}$) is 12% lower than that of the high-density a-IGZO film. Presumably, this result might be attributed to the physical movement of sputtered particles during the magnetron sputtering. The lower sputtering pressure and low O${}_{2}$ partial pressure could reduce the kinetic energy loss caused by the scattering and oxidation between the sputtered particles and the Ar, O${}_{2}$ gas, such that the sputtered particles have high kinetic energy. This effect increased the mobility of atoms at the surface of the film, thereby improving the structural relaxation, and ultimately leading to the formation of high-density a-IGZO films. This in turn reduced film defect density, which is essential for fabricating high-performance TFT devices.

To investigate the trap states in a-IGZO films in detail, we used the μ-PCD method to obtain the photoconductivity-response characteristics of a-IGZO films. In the μ-PCD measurement, laser irradiation activated excess carriers to increase the conductivity, resulting in a change in the microwave reflectivity of the sample, which is directly related to the change in the density of excess carriers. Generally, the photoconductivity will rapidly increase after the irradiation of laser light and is then saturated by trapping the carriers at defects. Therefore, the decay curves collected from μ-PCD contains three components: peak value, fast decay, and slow decay. The first component is the peak reflectivity signal which originated from the increased density of the photogenerated carriers through carrier generation and recombination processes during laser pulse irradiation, which is related to the density of the conduction band tail states [13]. In previous reports, the peak value has a strong correlation with the mobility of the TFTs [14]. The second component, or the fast decay, indicates the rapid recombination of the photogenerated carriers after laser pulse irradiation. This fast decay component is related to the recombination process through the deep level states. However, the lifetime of rapid decay is very short, so it is difficult to be observed directly. When the pulse width of the laser is large enough compared with the lifetime, the peak value is proportional to the lifetime. Therefore, for the evaluation of deep level traps, the μ-PCD uses the peak value, which can be quickly and accurately measured, rather than using the lifetime value of the fast decay [15]. Following the second component, a slow decay component appears with a decay constant in the order of microseconds. The mechanism of the slow decay component is attributed to the phenomenon that the laser-excited carriers are trapped at the localized states below the conduction band of the semiconductor thin film, and then de-trapped into the conduction band by thermal emission [16]. In general, the peak value and the slow decay time extracted from the μ-PCD curves are performed to evaluate carrier trap defects in semiconductors. As long as the peak value is higher and the decay time is shorter, the film quality will be better.

Figure 3 depicts the effects of O${}_{2}$ ratio, sputtering pressure, and annealing temperature on the μ-PCD curves, respectively. The results showed that with the change of deposition conditions, the peak value and decay time obtained from μ-PCD curves also changed. As shown in Figure 3a, we could observe that as the O${}_{2}$ ratio increased, the peak values dropped sharply. The a-IGZO films deposited in the absence of O${}_{2}$ exhibited a high peak value up to 579 mV. When the O${}_{2}$ ratio increases to 70%, the peak value dropped to 211 mV. It is worth noting that the peak value corresponded to the photogenerated carriers activated by the laser pulse irradiation, which means that the high O${}_{2}$ ratio in sputtering gas induced a higher density of deep-level defects, thereby reducing the number of the activated carriers. Besides, when the O${}_{2}$ ratio was 0, it exhibited a longer decay time. It is known that longer decay time means longer carrier lifetime, which indicates there are more trapped capture centers and defects in the a-IGZO films deposited without the incorporation of O${}_{2}$. As long as O${}_{2}$ was added into the sputtering atmosphere, the decay time was significantly shortened but varied weakly with increasing the O${}_{2}$ ratio. These results revealed that O${}_{2}$ was essential to healing the localized traps for the deposition of the high-quality a-IGZO film.

As shown in Figure 3b, when the pressure exceeded 1.33 Pa, the peak value of the μ-PCD curve was significantly reduced. In contrast, the relationship between the decay time and the sputtering pressure was weak. According to the peak values extracted from the μ-PCD curves, when the sputtering pressures were 0.13 Pa and 0.67 Pa, the corresponding peak values of the a-IGZO film were 546 mV and 561 mV, respectively. When the sputtering pressure increased to 1.33 Pa, the peak value quickly dropped to 302 mV. When the pressure continued to increase to 2.67 Pa, the peak value dropped to 266 mV. Based on the above analysis, we could infer that the lower peak value and longer decay time, on the one hand, might be due to the sufficient oxidation caused by the abundant oxygen, which in turn resulted in the lower carrier concentration. On the other hand, it might be ascribed to the traps induced by the sparse film structure.

We also discussed the effect of annealing temperature on film quality. Figure 3c shows the variation of μ-PCD curves collected from a-IGZO films annealed at different temperatures. The peak value of the μ-PCD curve increased with the increase of annealing temperature. Without annealing treatment, the peak value of the a-IGZO film was only about 9 mV. When the annealing temperature increased to 450 °C, the peak value rose to 561 mV. This result indicated that high-temperature annealing was beneficial to eliminate deep defects in the film, thereby stimulating more carriers and increasing carrier concentration. On the other hand, due to the extremely weak signal intensity of the unannealed IGZO film, the slow decay characteristic was hardly observable. When the annealing temperature increased from 250 °C to 350 °C, the decay time was almost constant. With further increasing the temperature to 450 °C, the decay time was remarkably shortened, indicating that a high temperature over 350 °C was required to eliminate the shallow-level traps in the a-IGZO films.

Finally, we fabricated a-IGZO TFTs under various conditions, and aimed to clarify the relationship between the film quality and the device performance. Figure 4 shows the transfer curves of these a-IGZO TFTs, where *V*${}_{G}$ swept from negative (−20 V) to positive (20 V) and back. Additionally, the electrical performance of these TFTs is summarized in Table 1. The saturation mobility ($\mu_{sat}$) and the threshold voltage (*V${}_{th}$*) were obtained from the curve of *I*${}_{D}^{1/2}$ versus the *V*${}_{G}$, which can be expressed as follows [17]:(1)$$I_{D}^{1/2}={\left(\frac{WC_{i}\mu_{sat}}{2L}\right)}^{1/2}\left(V_{G}-V_{th}\right)$$
where *C*${}_{i}$ is the gate capacitance per unit area of the insulator layer. The subthreshold swing (SS) was extracted as the inverse of the maximum slope of the curve of *Log I*${}_{D}$ versus the *V*${}_{G}$ [17]:(2)$$SS={\left[{\left(\frac{dLogI_{D}}{dV_{G}}\right)}_{max}\right]}^{-1}$$

The TFT manufactured under the deposition condition with an O${}_{2}$ ratio of 5%, a sputtering pressure of 0.67 Pa, and an annealing temperature of 450 °C had a saturation mobility of 8.4 cm${}^{2}$/Vs, a threshold voltage of 0.9 V, and a subthreshold swing of 0.16 V/dec. When the O${}_{2}$ ratio increased to 70%, $\mu_{sat}$ drops sharply to 2.7 cm${}^{2}$/Vs. This is highly consistent with the change of the peak values of a-IGZO films with the O${}_{2}$ ratio (Figure 3a). It is well known that oxygen deficiency in metal oxide semiconductors may act as deep-level electron traps and shallow donors [18]. Excessive oxygen may fill some of the oxygen vacancies and eventually reduce the carrier concentration. A similar trend was observed for the correlation between $\mu_{sat}$ and the sputtering pressure. That is, $\mu_{sat}$ decreased as the sputtering pressure increased. On the other hand, SS increaseds with the sputtering pressure. It is well known that SS is related to the total trap density, including the interface defects and the bulk trap density [19,20]. A higher SS value indicates a higher trap density in the a-IGZO channel layer. Therefore, the results are correlated to the film density and the photoconductivity property. It should be noted that the a-IGZO film deposited at 0.13 Pa showed the best film quality, but the TFTs prepared at 0.67 Pa exhibited the best transistor performance. This phenomenon may be attributed to the imperfect interface between a-IGZO and Al${}_{2}$O${}_{3}$. When the a-IGZO film was deposited at 0.13 Pa, the sputtered atoms had high kinetic energy due to less scattering of sputtering gas molecules. As a result, the sputtering atoms had a strong bombardment effect on Al${}_{2}$O${}_{3}$, resulting in an imperfect interface between a-IGZO and Al${}_{2}$O${}_{3}$, which hindered the transport of carriers, leading to a degraded TFT performance. In Figure 4c, when the annealing temperature increased, a dramatic change observed in the transfer curves was that the $\mu_{sat}$, *V*${}_{th}$, *SS* of the TFT were all greatly improved. In addition, we could observe that, under low-temperature annealing conditions, a-IGZO TFT exhibited obvious hysteresis, and as the annealing temperature increased, the hysteresis was significantly suppressed. In particular, for devices annealed at 450 °C, the hysteresis was almost not visible. Generally, the observed hysteresis effect was mainly related to the hole/electron traps in the semiconductors. Therefore, the suppressed hysteresis implies that a higher annealing temperature could effectively eliminate the traps in the film. This is consistent with the higher peak value and shorter decay time observed from the μ-PCD curves, as shown in Figure 3c. This result further shows that the film quality played a critical role in determining the device performance.

## 4. Conclusions

In summary, we systematically studied the effect of film quality on device performance by XRR and μ-PCD measurements. The results show that the film density, the peak value, and the decay time of the μ-PCD curve are essential parameters to determine the film quality. The changing trend of the film quality is highly consistent with the change of the transistor characteristics. The results show that a low oxygen ratio, a low sputtering pressure, and a high-temperature treatment are helpful to produce a dense, high-quality a-IGZO thin film with few defects. With these improvements, high-performance TFTs were demonstrated, with a saturation mobility of 8.4 cm${}^{2}$/Vs, a threshold voltage of 0.9 V, and a subthreshold swing of 0.16 V/dec.

## Figures and Tables

**Figure 1 nanomaterials-11-00522-f001:**
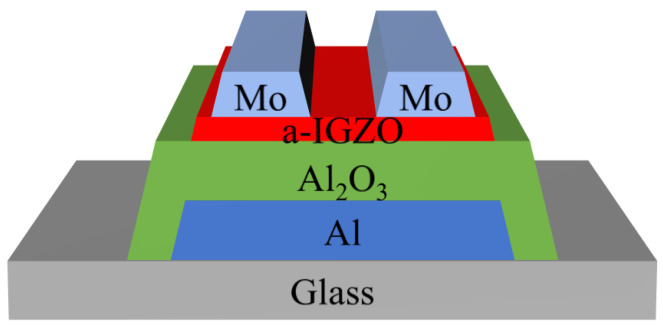
Schematic of the amorphous Indium-Gallium-Zinc Oxide (a-IGZO) thin-film transistors (TFTs).

**Figure 2 nanomaterials-11-00522-f002:**
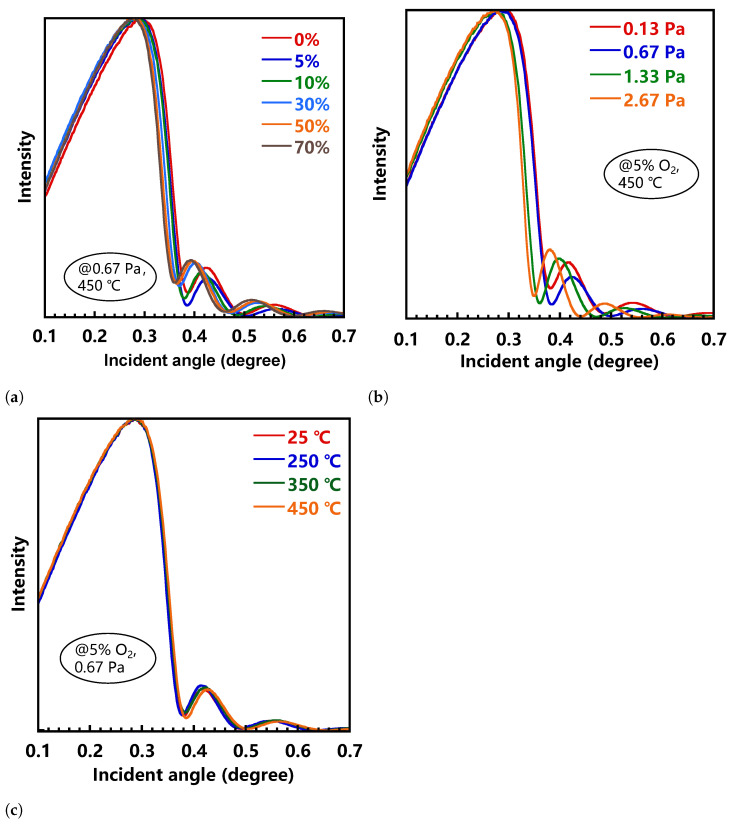
X-ray Reflectivity (XRR) curves collected from a-IGZO films prepared at (**a**) different O${}_{2}$/(Ar+O${}_{2}$) ratio (for 0.67 Pa, 450 °C), (**b**) sputtering pressure (for 5% O${}_{2}$, 450 °C), and (**c**) annealing temperature (for 5%O${}_{2}$, 0.67 Pa).

**Figure 3 nanomaterials-11-00522-f003:**
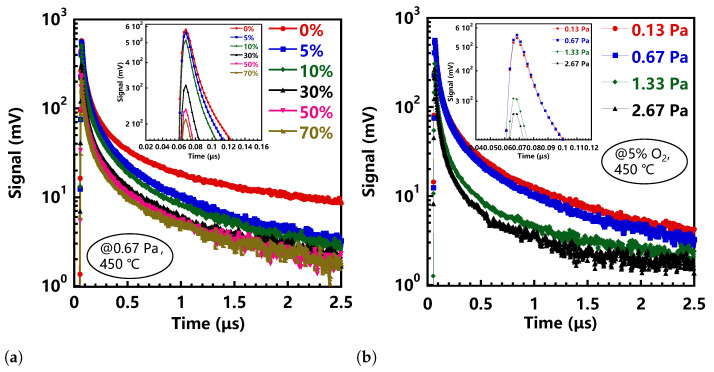
Microwave Photoconductivity Decay curves obtained from a-IGZO films prepared at (**a**) different O${}_{2}$/(Ar+O${}_{2}$) ratio(for 0.67 Pa, 450 °C), (**b**) sputtering pressure (for 5% O${}_{2}$, 450 °C), and (**c**) annealing temperature (for 5%O${}_{2}$, 0.67 Pa).

**Figure 4 nanomaterials-11-00522-f004:**
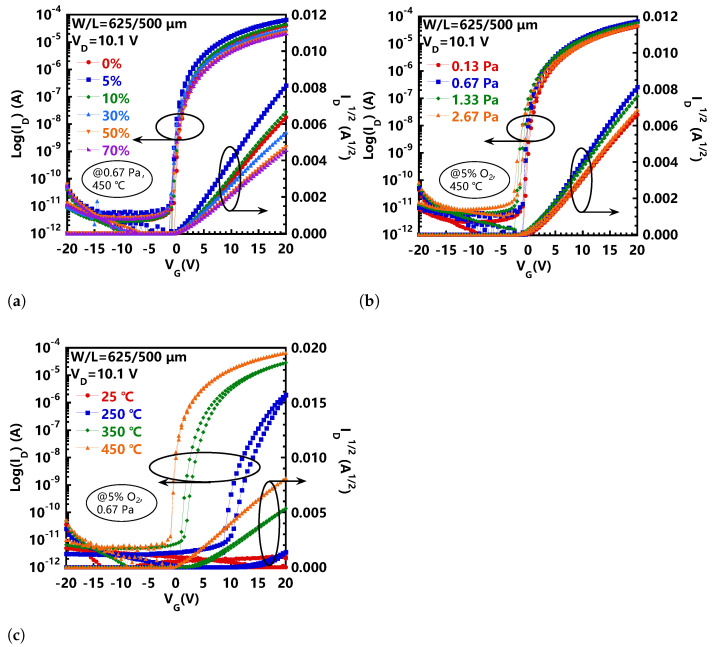
Transfer characteristics of TFTs using a-IGZO films deposited at (**a**) different O${}_{2}$/(Ar+O${}_{2}$) ratio, (for 0.67 Pa, 450 °C), (**b**) sputtering pressure (for 5% O${}_{2}$, 450 °C), and (**c**) annealing temperature (for 5% O${}_{2}$, 0.67 Pa).

**Table 1 nanomaterials-11-00522-t001:** Variation of the performance for both a-IGZO TFTs.

O${}_{2}$ Ratio	Pressure (Pa)	T${}_{\mathrm{anneal}}$ (°C)	V${}_{\mathrm{th}}$ (V)	$\mu_{\mathrm{sat}}$ (cm${}^{2}$/Vs)	*SS* (V/dec)
0%	0.67	450	2.0	5.7	0.14
5%	0.13	450	1.7	5.9	0.18
5%	0.67	—	—	—	—
5%	0.67	250	14.4	2.6	0.56
5%	0.67	350	4.5	5.1	0.34
5%	0.67	450	0.9	8.4	0.16
5%	1.33	450	0.6	6.9	0.33
5%	2.67	450	1.1	5.8	0.46
10%	0.67	450	1.7	6.0	0.19
30%	0.67	450	1.1	3.9	0.20
50%	0.67	450	1.8	3.0	0.19
70%	0.67	450	1.5	2.7	0.29

## Data Availability

The data presented in this study are available on request from the corresponding author. The data are not publicly available due to privacy

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
