# Peer review of "Study of the Correlation between the Amorphous Indium-Gallium-Zinc Oxide Film Quality and the Thin-Film Transistor Performance"

_nanomaterials, 2021, doi:10.3390/nano11020522_

Round 1

Reviewer 1 Report

Replace the old unit Torr by Pascal in all places.

Fig 2: What about the measurement error? Differences are small and maybe in the range of measurement error.

Fig 3: the peak values cannot be seen in the figures

Fig. 4b: According your arguments, 1 mTorr should be the best, but you indicate 5 mTorr to be best

Table 1: What about the measurement errors? How many runs of individual parameters did you run? As the differences between 5 mT and 10mT are small, and the same for 0% O2, 5%O2 and 10%O2 are very small concerning µ, Vth and SS (SS is even better for 0% O2), I assume the results may differ from run to run. Are the results really as strongly related to the small process parameter variations?

Reviewer 2 Report

In this paper, the authors investigated relationship between a-IGZO film quality and TFT performance. The contents include some interesting and useful information. I will give the following comments.
(1) You wrote "... Al2O3 gate insulator was formed by an anodizing process". So, why the parts apart from Al are covered with Al2O3 in Fig. 1 ?
(2) You wrote "The optimal density can be as haigh as 6.0g/cm3". How you can get this density ? Could you explain how to calculate the density from Fig. 2.

Reviewer 3 Report

The recommendations are enclosed.

Round 2

Reviewer 1 Report

Thank you for clearifying all points.